# The Pathophysiological Role of Thymosin β4 in the Kidney Glomerulus

**DOI:** 10.3390/ijms24097684

**Published:** 2023-04-22

**Authors:** William J. Mason, Elisavet Vasilopoulou

**Affiliations:** 1Developmental Biology and Cancer Programme, UCL Great Ormond Street Institute of Child Health, London WC1N 1EH, UK; w.mason@ucl.ac.uk; 2Comparative Biomedical Sciences, The Royal Veterinary College, London NW1 0TU, UK

**Keywords:** thymosin β4, glomerulus, podocyte, macrophage, chronic kidney disease

## Abstract

Diseases affecting the glomerulus, the filtration unit of the kidney, are a major cause of chronic kidney disease. Glomerular disease is characterised by injury of glomerular cells and is often accompanied by an inflammatory response that drives disease progression. New strategies are needed to slow the progression to end-stage kidney disease, which requires dialysis or transplantation. Thymosin β4 (Tβ4), an endogenous peptide that sequesters G-actin, has shown potent anti-inflammatory function in experimental models of heart, kidney, liver, lung, and eye injury. In this review, we discuss the role of endogenous and exogenous Tβ4 in glomerular disease progression and the current understanding of the underlying mechanisms.

## 1. Introduction

The kidney glomerulus is a specialised capillary bed that constitutes the filtration unit of the kidney. The glomerular filtration barrier consists of endothelial cells, the glomerular basement membrane, and epithelial podocytes and is supported by mesangial cells. Glomerular injury is regarded as damage to one or more components of the glomerulus that hinders its ability to perform ultrafiltration of the blood. This can manifest in a diverse range of phenotypes, such as mesangial matrix expansion, collapse of capillary loops, podocyte foot process effacement, podocyte detachment, thickening of the glomerular basement membrane, macrophage infiltration, and glomerulosclerosis [1,2,3,4,5]. Glomerular disease can lead to albuminuria and declining renal function and is a major contributor to end-stage kidney disease [1].

Thymosin β4 (Tβ4), a peptide encoded by the *TMSB4X* gene on the X chromosome [6], is pleiotropic and has cellular functions that are relevant in glomerular disease. As the major G-actin-sequestering protein in mammalian cells, Tβ4 maintains cytoskeletal dynamics [7] and modulates cell morphology and cell motility [8]. In addition, Tβ4 promotes cell differentiation, [9] survival [10], and angiogenesis [11] and inhibits inflammation [12] and fibrosis [13]. As a result, Tβ4 has been shown to have a protective role in a number of animal disease models including myocardial infarction [14], stroke [15], dry eye syndrome [16], and pulmonary fibrosis [17]. Here, we discuss the current understanding of the role of Tβ4 in the kidney glomerulus. We focus on two cell types that express Tβ4 in the kidney: podocytes, which form an integral part of the glomerular filtration barrier and have a complex shape maintained by their actin cytoskeleton [5], and macrophages, which are involved in inflammatory processes that drive disease progression following glomerular injury [18].

## 2. Podocytes

### 2.1. Podocytes and the Actin Cytoskeleton

Podocytes are specialised epithelial cells with a unique shape that is integral to the function of the glomerular filtration barrier [19]. Primary processes extend from the podocyte cell body and branch into foot processes that wrap around the glomerular capillaries and interdigitate with foot processes from adjacent podocytes [20]. The gap between the foot processes, the slit diaphragm, forms a size- and charge-selective barrier that allows the passage of small molecules, while large blood components, such as albumin and other plasma proteins and erythrocytes, are retained in the blood [21].

A highly regulated network of filamentous actin (F-actin) is critical for maintaining podocyte shape and glomerular filtration [22]. Early reports using immunogold-labelling electron microscopy and more recent studies utilising super-resolution microscopy have revealed that healthy podocytes contain non-contractile F-actin fibres localised to the foot processes, with a cortical network parallel to the plasma membrane and a thick centrally running F-actin cable (Figure 1a) [23,24]. Contractile F-actin is localised to primary processes and cell bodies, where it creates a shell in the cytoplasm adjacent to the cell membrane, supporting the unique structure of the podocyte [23,24]. The basal membrane of the foot processes adheres to the glomerular basement membrane through focal adhesions. These focal adhesions are mostly composed of a3β1 integrin and aβ dystroglycan molecules, which bind to laminin and agrin in the glomerular basement membrane and are linked to the cytoskeleton through linker proteins [25,26,27,28]. Chronic kidney disease (CKD) is associated with disorganisation of the podocyte actin cytoskeleton, leading to foot process effacement, detachment from the glomerular basement membrane, and podocyte loss (Figure 1b) [23]. These structural changes result in impaired renal filtration demonstrated by leakage of plasma proteins, such as albumin, in the urine. Albuminuria is a hallmark of glomerular disease, irrespective of the underlying aetiology [1]. Therefore, maintaining the integrity of the podocyte cytoskeleton is critical to prevent albuminuria, preserve glomerular function, and halt CKD progression. In this section, we review the current evidence on the role of Tβ4 in podocytes.

### 2.2. Tβ4 Expression in Podocytes

Several studies have demonstrated Tβ4 expression in podocytes. The mRNA transcript for *Tmsb4x* was detected in developing glomeruli in embryonic day 16 [29], embryonic day 18, and in mature glomeruli in kidneys from 8-week-old mice by in situ hybridisation [30] with a localisation pattern indicative of expression in podocyte cells. These findings were corroborated by microarray analysis that identified *Tmsb4x* as a transcript enriched in adult mouse podocyte cells [31] and more recently by analysis of single-cell RNA sequencing (scRNAseq) datasets [32,33]. Tβ4 expression has also been demonstrated at the protein level in developing and mature mouse glomeruli by immunohistochemistry, and the localisation of Tβ4 peptide in podocytes has been identified by co-localisation with the podocyte markers nephrin and nestin [29]. *Tmsb4x* transcripts were also detected in both developing and mature zebrafish podocytes by scRNAseq, suggesting that podocyte expression of Tβ4 is conserved in different species. Interestingly, this study identified a switch in aged podocytes from *Tmsb4x* to *Tmsb1*, another member of the beta thymosin family found in zebrafish and other bony fish [34]. *TMSB4X* transcripts are detected in lysates from adult human glomeruli [29]; however, there is little information about the expression of Tβ4 specifically in podocytes in humans. Analysis of scRNAseq datasets from isolated human glomeruli could provide insight into the levels of *TMSB4X* in human podocytes. These studies demonstrate that Tβ4 is expressed in podocyte cells and therefore could play a role in maintaining podocyte structure and function.

Given the presence of Tβ4 in podocytes and its role in actin binding [7], it is important to examine whether glomerular injury affects podocyte Tβ4 levels. Initial studies focused on assessing Tβ4 levels in whole kidney or glomerular extract lysates. A proteomic study [35] utilised the rat remnant kidney model, which involves unilateral nephrectomy of the right kidney and ligation of the branches of the left renal artery, resulting in 5/6 renal ablation, glomerulosclerosis, and interstitial fibrosis. Tβ4 levels were increased threefold in sclerotic versus normal glomeruli. Immunostaining localised Tβ4 predominately to endothelial cells with no staining detected in podocytes [35]. In contrast, *Tmsb4x* mRNA levels were unchanged in whole kidney lysates from mice with nephrotoxic nephritis [29], an immune-mediated model of glomerular injury that replicates some of the pathologic features of human crescentic glomerulonephritis [36], compared with healthy controls. In this model, Tβ4 was expressed in podocytes in the injured kidney. *TMSB4X* mRNA levels were also unchanged in the glomerular lysates from kidney biopsy specimens obtained from patients with either rapidly progressive glomerulonephritis or lupus nephritis compared with living donor control kidneys [29]. Recent technological advances have enabled interrogation of mRNA levels at the single-cell level, thus allowing quantification of *Tmsb4x* expression in a cell-specific manner. A study by Chung et al. generated single-cell transcriptomic profiles of the glomerulus from healthy mice and mice with different types of glomerular injury, including mice with nephrotoxic nephritis; mice injured with Adriamycin, a chemotherapeutic drug which replicates some of the features of human focal segmental glomerulosclerosis; and BTBR *ob/ob* (*Lepr^Ob/Ob^*) mice with diabetic kidney disease [32]. Analysis of these scRNAseq datasets revealed that the podocyte expression of *Tmsb4x* was significantly reduced in all these disease models compared with *Tmsb4x* levels in the podocytes of healthy control mice [33]. In addition, the expression of another member of the beta-thymosin family which has been previously described in podocytes [29], *Tmsb10*, was also downregulated in injured compared with healthy podocytes. Similar observations were made in vitro where injury of mouse immortalised podocytes with Adriamycin resulted in a reduction in the mRNA levels of *Tmsb4x* and *Tmsb10* in a dose-dependent manner [33]. These findings highlight that podocyte injury is associated with reduced expression of *Tmsb4x* across multiple mouse models of glomerular disease and that these changes are not accurately reflected when assessing the expression of *Tmsb4x* in glomerular or whole kidney extracts.

### 2.3. The Role of Tβ4 in Podocyte Pathophysiology

Mice with global deletion of *Tmsb4x* have been used to elucidate the role of endogenous Tβ4. Despite the association of podocyte injury with reduced *Tmsb4x* expression, kidneys from male *Tmsb4x^−/y^* and *Tmsb4x^+/y^* mice had similar glomerular morphology assessed by PAS staining, podocyte structure assessed by electron microscopy [29], and protein levels of nephrin, a protein which is localised in podocytes in the kidney and is an essential component of the slit diaphragm [37]. In agreement with these findings, *Tmsb4x^−/y^* mice had normal levels of urinary albumin, plasma creatinine, and blood urea nitrogen [29,37], indicating that there are no defects in the glomerular filtration barrier, and renal function is maintained. These findings suggest that endogenous Tβ4 is dispensable for healthy podocyte development, mature ultrastructure, and function in vivo.

The pathophysiological role of endogenous Tβ4 was determined by further experiments in murine models of glomerular injury. Two distinct mouse models were examined: nephrotoxic nephritis [29], a model of autoimmune crescentic glomerulonephritis [36], and Angiotensin-II (Ang-II)-induced hypertension causing end-organ damage including glomerular injury [37]. In both cases, Tβ4 deficiency was shown to accelerate disease progression. Nephrotoxic nephritis was characterised by high levels of albuminuria and changes in glomerular morphology including glomerulosclerosis, adhesion of the glomerular tuft to the surrounding Bowman’s capsule, and epithelial hyperplasia lesions in wild-type mice 21 days post disease induction. These features were exacerbated in *Tmsb4x^−/y^* mice, which had significantly higher levels of albuminuria as well as increased plasma creatinine and blood urea nitrogen and a greater degree of glomerular injury as assessed by periodic acid Schiff (PAS) staining. Further experiments showed that nephrotoxic nephritis resulted in loss of podocytes, identified by immunofluorescent staining for Wilms tumour 1 (WT1), from the glomerular tuft in *Tmsb4x^−/y^* mice. This was accompanied by increased WT1^+^ nuclei in the parietal epithelium lining the Bowman’s capsule, raising the possibility that Tβ4 deficiency results in aberrant migration of injured podocytes from the glomerular tuft to the Bowman’s capsule [29]. In a separate model, infusion of Ang-II raised systolic blood pressure and resulted in low levels of albuminuria in wild-type mice. Albuminuria was significantly higher in *Tmsb4x^−/y^* mice despite having similar levels of hypertension as wild-type mice. Podocyte number was not assessed directly in this study, but the protein levels of nephrin were lower in kidneys from hypertensive *Tmsb4x^−/y^* compared with *Tmsb4x^+/y^* mice [37]. Reduced levels of nephrin could be explained by either downregulation of nephrin expression or loss of podocyte cells. Overall, these studies demonstrate that endogenous Tβ4 has a protective role maintaining the integrity of the glomerular filtration barrier following injury and that this is at least partly mediated by preventing podocyte injury.

The direct effect of endogenous Tβ4 on podocyte cells was demonstrated by in vitro experiments. The viability of immortalised mouse podocytes [38] was unaffected after >90% knockdown of *Tmsb4x* by siRNA. However, lack of Tβ4 resulted in reorganisation of the F-actin cytoskeleton from predominantly cortical F-actin to cytoplasmic stress fibres and increased podocyte migration in a scratch wound assay. In addition, downregulation of Tβ4 increased RhoA activity, a master regulator of F-actin organisation, stress fibre formation, and migratory control [29], offering mechanistic explanation to these results. These findings demonstrate that endogenous Tβ4 regulates the podocyte actin cytoskeleton with downstream functional effects which may explain why in vivo podocytes that lack Tβ4 are more susceptible to injury.

The findings that Tβ4 has a protective role in the glomerulus and that podocyte Tβ4 expression is reduced in murine models of glomerular disease raise the possibility that treatment with exogenous Tβ4 may slow glomerular disease progression. Initial evidence that exogenous Tβ4 may prevent glomerular injury came from a study using the *KK Cg-Ay* mouse model of type 2 diabetes [39]. Twelve-week-old *KK Cg-Ay* mice were injected with either saline or 100 ng/10 g/day Tβ4 intraperitoneally for three months. Tβ4 injections suppressed albuminuria by 33% compared with saline-injected littermates [40]. Tβ4 treatment also reduced blood glucose levels in this study, which could have contributed to the improvement in glomerular filtration. The effects of Tβ4 on podocytes were not assessed in this study. It is important to note that exogenous Tβ4 administration in wild-type mice did not alter renal function [40], providing evidence that exogenous Tβ4 does not have adverse effects in the kidney.

Subsequently, the mouse model of Adriamycin nephropathy, which primarily targets podocytes, was used to assess the effect of exogenous Tβ4 on podocyte injury [33] (Figure 2). A particular strength of this study was using gene therapy to achieve sustained systemic upregulation of Tβ4. Previous work showed that after intravenous injection of Tβ4 peptide in mice, the plasma concentration of Tβ4 returns to basal levels within 6 h [41]. In this study, wild-type BALB/c mice were injected intravenously with adeno-associated virus (AAV) 2/7 encoding *Tmsb4x.* Tβ4 protein levels were assessed five weeks later, and it was shown that gene therapy with Tβ4 resulted in transduction of liver cells followed by secretion of Tβ4 and a twofold increase in Tβ4 plasma levels in mice injected with AAV-*Tmsb4x* compared with controls injected with AAV-*LacZ*. Validation experiments in *Tmsb4x^-/y^* mice showed that AAV-delivered Tβ4 was internalised by podocyte cells, demonstrating that this is an effective strategy to systemically deliver Tβ4 to podocyte cells over a prolonged period [33]. A preventative strategy was used, where mice were intravenously injected with Adriamycin (10 mg/kg) three weeks after AAV administration. Gene therapy with Tβ4 prevented Adriamycin-induced albuminuria, with urinary albumin concentration comparable to the healthy control mice, demonstrating that exogenous Tβ4 protects the glomerular filtration barrier. In addition, Tβ4 administration prevented the Adriamycin-induced loss of podocyte cells, as measured by counting WT-1+ nuclei, indicating that prevention of podocyte injury is potentially a mechanism by which Tβ4 prevents albuminuria [33]. There was no change in the total amount of F-actin within podocytes, identified by synaptopodin staining, but the maximum resolution achievable by this method did not allow detailed assessment of F-actin organisation. However, injury of mouse immortalised podocytes with Adriamycin in vitro resulted in a shift from cortical F-actin to unorganised F-actin in a dose-dependent manner. Treatment with synthetic Tβ4 peptide (100 ng/mL) prevented the F-actin disorganisation induced by Adriamycin [33], suggesting that the protective role of exogenous Tβ4 in glomerular disease may be mediated via protection of the podocyte cytoskeleton from injury. Novel techniques utilising super-resolution microscopy to examine podocyte F-actin in vivo [23,42] could help determine whether Tβ4 maintains podocyte F-actin organisation in vivo.

Collectively, the available literature illustrates how endogenous and exogenous Tβ4 protects the glomerular filtration barrier in pathological settings, likely by preserving the podocyte cytoskeleton and thus maintaining podocyte function in the glomerular tuft. However, the molecular mechanisms that underpin the protective effect of Tβ4 on podocytes have not been investigated in detail. The F-actin cytoskeleton links focal adhesion complexes to the foot processes maintaining anchorage to the glomerular basement membrane [43]; therefore, it is possible that Tβ4 prevents podocyte loss by preserving the podocyte cytoskeleton and thus the localisation and adherence of podocytes to the glomerular tuft. In the epidermis, Tβ4 depletion results in planar cell polarity (PCP) defects due to defective G-actin and F-actin distribution and abnormal structural organisation and stability of adherens junctions. [44] PCP is involved in podocyte maturation during kidney organogenesis and disease [45]. Depletion of Van Gogh-like 2 (Vangl2), a core PCP protein, in podocytes exacerbated albuminuria in the nephrotoxic nephritis mouse model [46,47]. PCP proteins may therefore mediate the effects of Tβ4 on podocyte cytoskeletal organisation and function. Sequestration of G-actin by Tβ4 promotes the release of myocardin-related transcription factor A (MRTFA) from G-actin and its translocation to the nucleus where it binds to serum response factor (SRF) [48]. The MRTFA-SRF complex regulates genes linked to cytoskeletal dynamics and migration [49]; therefore, the effects of Tβ4 on the podocyte cytoskeleton could be an indirect effect of Tβ4 promoting MRTFA-SRF signalling. Tβ4 forms a complex with integrin-linked kinase (ILK) and PINCH, resulting in the release of Akt [10]. This signalling pathway promotes cell survival, migration, and adhesion [8,50], presenting another potential mechanism mediating the protective effect of Tβ4 on podocytes.

The protective effect of Tβ4 on podocytes may also be mediated by its metabolite, the tetrapeptide N-acetyl-seryl-aspartyl-lysyl-proline (Ac-SDKP), which is generated by successive cleavage of Tβ4 by the enzymes meprin-α [51] and prolyl oligopeptidase (POP) [52] and is degraded by the angiotensin-I-converting enzyme [53]. AcSDKP has shown beneficial effects in a number of experimental models of kidney disease, most prominently showing anti-inflammatory and anti-fibrotic properties [54,55]. AcSDKP administration also improves albuminuria in murine models of hypertensive nephropathy [56,57,58], lupus nephritis [59,60], and nephrotoxic nephritis [61], suggesting that it protects the glomerular filtration barrier. AcSDKP has a beneficial effect in models of diabetic nephropathy where it improves kidney fibrosis [62,63,64]; however, its effect on glomerular filtration is less clear. One study reported that AcSDKP had no effect on albuminuria induced by streptozotocin administration in rats [62]. In contrast, administration of AcSDKP in mice with streptozotocin-induced diabetes improved albuminuria and ameliorated podocyte foot process effacement [63]. Minor improvement of albuminuria by AcSDKP was reported by a study assessing its efficacy in diabetic db/db mice [64]. POP inhibitors could be utilised to determine whether the beneficial effect of Tβ4 on podocyte function is mediated to an extent via its metabolite, AcSDKP. Since AcSDKP does not include the G-actin-binding domain of Tβ4 [65,66], any therapeutic benefits observed would be independent of G-actin sequestering activity.

Overall, these studies demonstrate that both endogenous and exogenous Tβ4 protect the podocyte cytoskeleton, prevent podocyte loss, and maintain the integrity of the glomerular filtration barrier during injury. However, it is currently not known whether Tβ4 can reverse podocyte injury, and therefore further studies administering Tβ4 after the onset of glomerular injury could be employed as a more clinically relevant approach to assess its efficacy in improving the progression of established glomerular disease. In addition, studies utilising human podocytes can provide evidence for the relevance of Tβ4 in human glomerular disease.

## 3. Macrophages

### 3.1. Macrophage Infiltration in Glomerular Disease

Glomerular injury is often accompanied by an inflammatory response characterised by leukocyte infiltration and subsequent fibrosis which correlates with declining renal function [18]. Inflammation plays a major role in the progression of many kidney disorders including glomerulonephritis [18], diabetic nephropathy [67], and hypertensive nephropathy [68]. Renal inflammation is driven by cytokine release and expression of adhesion molecules by renal cells that promote monocyte infiltration and accumulation of macrophages in the kidney [69]. Renal macrophages consist of distinct subpopulations of different origin (embryo-derived and bone-marrow-derived) and divergent functions that may exacerbate kidney injury or conversely contribute to tissue repair [70,71,72,73,74,75,76,77,78,79]. Depletion of macrophages or inhibition of cytokines that drive macrophage recruitment attenuates glomerular injury in experimental models of glomerulonephritis [80], lupus nephritis [81,82], and diabetic nephropathy [83,84,85,86]. Therefore, therapies that target renal macrophage recruitment and activation have the potential to slow the progression of glomerular disease. Tβ4 reduces inflammation in experimental models of corneal wounds [87,88,89], lung disease [17], chronic granulomatous disease, and [90] inflammatory bowel disease [91]. In this section, we discuss the effects of Tβ4 on macrophage infiltration in the context of glomerular disease. The anti-inflammatory role of Tβ4 and AcSDKP in kidney disease that primarily targets the tubulointerstitium have been reviewed previously [54,55,92,93].

### 3.2. The Role of Tβ4 in Kidney Macrophage Accumulation following Glomerular Injury

Initial data from the rat remnant kidney model suggested that Tβ4 is not expressed in kidney macrophages identified by ED1 staining [35]. Subsequently, Tβ4 staining was identified in kidney macrophages identified by F4/80 in the mouse model of nephrotoxic nephritis [29]. This discrepancy may be due to differences between species, type and stage of CKD, or the antibodies available to detect Tβ4. Additionally, Tβ4 expression in macrophages may differ according to macrophage subtype and activation state which was not assessed in these studies that used pan-macrophage markers.

Endogenous Tβ4 inhibits macrophage accumulation in the kidney cortex after injury, as demonstrated by both the studies that utilised Tβ4 knockout mice. In the mouse nephrotoxic nephritis model, the number of macrophages identified by F4/80 immunostaining was increased at the early stage of the disease (day 7) but did not differ between *Tmsb4x^−/y^* and *Tmsb4x^+/y^* mice. In contrast, at the later stage of the disease (day 21) there were more macrophages in both the glomerular tuft and the peri-glomerular region in *Tmsb4x^−/y^* compared with *Tmsb4x^+/y^* mice. This was accompanied by increased collagen IV and α-smooth muscle actin (SMA) staining in these regions, indicating increased fibrosis [29]. In the mouse model of Ang-II-induced hypertension, the infiltration of macrophages, identified by immunostaining for Cd68, in the renal cortex was exacerbated in *Tmsb4x^−/y^* compared with *Tmsb4x^+/y^* mice. This was accompanied by increased expression levels of Intercellular Adhesion Molecule 1 (ICAM-1), a pro-inflammatory molecule; collagen content assessed by picrosirius red staining; and α-SMA, quantified by Western blotting [37]. These results suggest that endogenous Tβ4 promotes the resolution of inflammation following glomerular injury and prevents subsequent fibrosis.

The mechanisms that mediate the anti-inflammatory and anti-fibrotic effects of Tβ4 in glomerular disease are currently unclear. Tβ4 may regulate the production of cytokines and growth factors by injured glomeruli to limit monocyte infiltration and in situ macrophage proliferation. Alternatively, Tβ4 may act directly on macrophages to modulate their function. It is also currently not known if the anti-inflammatory role of Tβ4 in glomerular disease is dependent on its G-actin sequestering activity. Previous work has shown that Tβ4 inhibits tumour necrosis factor α (TNF-α)-induced nuclear factor κB (NF-κB) activation in human corneal cells independently of its G-actin sequestering activity [94]. In addition, Tβ4 metabolites may also play a role. Tβ4 sulfoxide disperses macrophages at the site of injury [95], and AcSDKP has anti-inflammatory and anti-fibrotic function in the kidney [54,55,92,93]. The inflammatory response has been associated with increased fibrosis in the kidney [96], and therefore the anti-fibrotic effects of Tβ4 in glomerular disease may be a downstream effect of its anti-inflammatory function. In addition, Tβ4 and Ac-SDKP may inhibit fibrosis in glomerular disease by inhibiting transforming growth factor β (TGF-β) and plasminogen activator inhibitor-1 (PAI-1) signalling, as has been shown in a model of renal interstitial fibrosis [97]. This pathway is also implicated in glomerular injury, where TGF-β stimulates mesangial cell extracellular matrix production [98] and PAI-1 synthesis, thus promoting matrix accumulation [99].

Given these findings, it is interesting to speculate whether administration of exogenous Tβ4 may also be effective in reducing macrophage accumulation in glomerular disease. The model of Adriamycin nephropathy that has been used to assess the effect of gene therapy with Tβ4 on glomerular disease does not result in inflammation within the time frame studied. This was illustrated by the low numbers of macrophages present in the renal cortex, which were similar in the mice that received Adriamycin and control mice that were administered saline [33]. A number of studies have demonstrated anti-inflammatory and anti-fibrotic effects of the Tβ4 metabolite AcSDKP in glomerular disease. In a rat model of nephrotoxic nephritis, AcSDKP administration reduced the renal expression of pro-inflammatory genes (ICAM-1, interleukin-1β, monocyte chemoattractant protein (MCP)-1, and TNF-α) and inhibited macrophage infiltration in both the glomerulus and the tubulointerstitium [61]. Similar results were obtained in rat and mouse models of hypertensive nephropathy, where AcSDKP administration reduced macrophage infiltration and collagen content in the renal cortex [56,57,58]. In a mouse model of lupus nephritis, AcSDKP treatment reduced macrophage infiltration in the glomerulus and tubulointerstitium, lowered renal levels of chemokines and cytokines (complement C5-9, RANTES, MCP-5, and ICAM-1), and reduced collagen content assessed by picrosirius red staining [59]. These studies demonstrate that AcSDKP has both anti-inflammatory and anti-fibrotic effects in the injured glomerulus. Further work using models of immune-mediated glomerular disease is needed to assess the anti-inflammatory potential of the parent peptide Tβ4 following glomerular injury and the molecular and cellular mechanisms involved.

## 4. Conclusions

In summary, Tβ4 has a protective role in glomerular disease as demonstrated in a range of experimental models. This effect is mediated by the ability of Tβ4 to protect the actin cytoskeleton of podocytes, thus preventing changes in the podocyte shape and podocyte loss and maintaining the integrity of the glomerular filtration barrier. At the same time, Tβ4 inhibits the accumulation of macrophages in the renal cortex, exerting an anti-inflammatory effect which is likely to slow disease progression as indicated by the concomitant reduction in fibrosis (Figure 3). It is currently not known whether Tβ4 affects other glomerular cells, either directly or indirectly, by regulating paracrine signalling from podocytes and infiltrating macrophages.

It is unclear whether the effects of Tβ4 on podocytes and macrophages are distinct, or whether Tβ4 administration modulates signalling pathways that drive inflammation in response to podocyte injury. Irrespective of the underlying cellular and molecular mechanisms, the studies conducted to date have demonstrated that Tβ4 can inhibit the development and severity of glomerular disease. More studies are required to explore the therapeutic potential of exogenous Tβ4 in established glomerular disease to assess whether it can halt or reverse disease progression, which will strengthen the evidence for using Tβ4 as a novel treatment to reduce the morbidity and mortality associated with glomerular disease. Further studies are also needed to determine the relevance of Tβ4 and its therapeutic potential in human glomerular diseases.

## Figures and Tables

**Figure 1 ijms-24-07684-f001:**
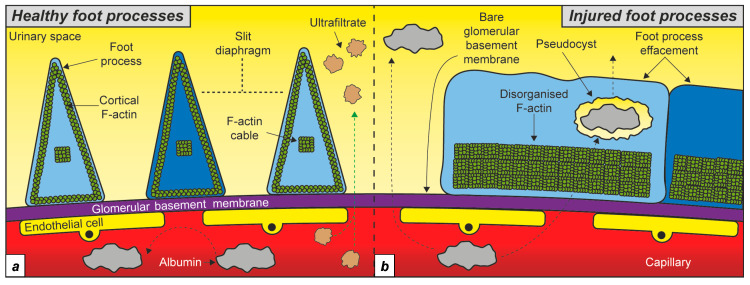
Podocyte F-actin in health and disease. (**a**) Schematic depicting the cortical F-actin and central F-actin cable in healthy foot processes. (**b**) Podocyte injury is associated with foot process effacement and disorganised F-actin running adjacent to the glomerular basement membrane. Different shades of blue indicate neighbouring podocyte cells.

**Figure 2 ijms-24-07684-f002:**
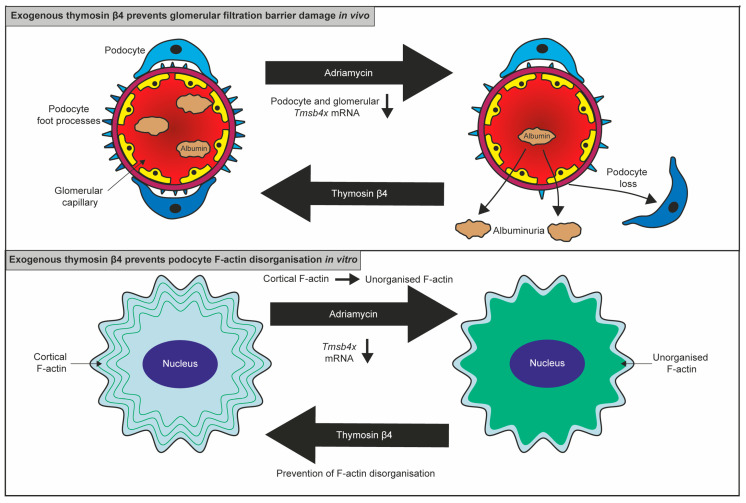
The protective effect of exogenous Tβ4 on podocytes. Adriamycin administration caused a reduction in podocyte *Tmsb4x* expression in vivo and in vitro. Adriamycin administration in mice resulted in podocyte loss and albuminuria which was prevented by Tβ4 administration. In cultured podocytes, Adriamycin induced actin cytoskeletal disorganisation which was prevented by treatment with exogenous Tβ4.

**Figure 3 ijms-24-07684-f003:**
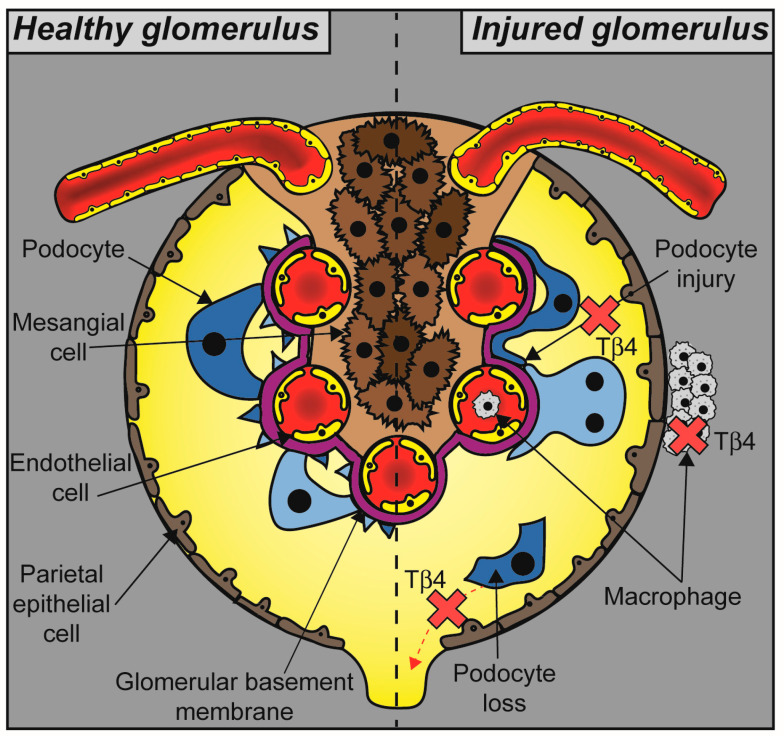
The role of Tβ4 in glomerular pathophysiology. Glomerular disease results in podocyte injury, foot process effacement, and detachment from the glomerular basement membrane leading to defective filtration and albuminuria. Accumulation of macrophages in the glomerular tuft and the periglomerular area is part of an inflammatory response that leads to fibrosis and accelerates disease progression. Tβ4 protects podocytes from injury and inhibits macrophage accumulation, thus maintaining glomerular function (indicated by red cross marks).

## Data Availability

Not applicable.

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
