# Peer review of "The Pathophysiological Role of Thymosin β4 in the Kidney Glomerulus"

_ijms, 2023, doi:10.3390/ijms24097684_

Round 1

Reviewer 1 Report

In the manuscript, Mason. et al. sumarized the role of Tβ4 in podocyte and macrophage.  This is a informative review. However, several aspects can be improved. 

Point 1. In line 35, the phrase "improve disease progression" means an adverse effect to exacerbate the disease. But the original literatures show a protective role. 

Point 2. Part 2.3 is too long, I would suggest to make the molecular mechanisms a separate part. 

Point 3. The molecular mechanisms discussed in Part 2.3 are not completely included in Figure 2. 

Point 4. Figure 3 contains too much redundant information, which conceals the leading role of Tβ4. 

Point 5. As shown in Figure 3, a glomerulus is composed of several different kinds of cells, it is better to discuss the role of Tβ4 in other cells as well.

Reviewer 2 Report

This review article is interesting, provides outstending data about the pathophysiological role of thymosin B4 in different glomerular diseases, based on plenty cited literature.

1. In introduction section, authors should use different frase instead of inflammatory lung disease. Also, frase dry eye should be replaced (line 36).

2. In section 2.3, line 23, symbol for meprin alpha should be corrected.

3. Authors should try to explaine, in more datil, potential role of tymosin B4 on macrophage function in glomerular injury. In this section, this is explained too generally, and it is not in contrast with previously extraordinary explained role of tymosin B4 in podocyte pathophysiology.

Reviewer 3 Report

This manuscript offers an extensive review of thymosin β4 (Tβ4) and its role in glomerular disease progression, focusing on the underlying mechanisms involving podocytes and macrophages. The authors discuss Tβ4's potential role in maintaining podocyte cytoskeleton integrity and modulating inflammation in macrophages. The review also investigates the effects of glomerular injury on Tβ4 expression and its involvement in podocyte pathophysiology. The authors highlight the protective role of Tβ4 in cases of glomerular injury, as well as the potential therapeutic benefits of exogenous Tβ4 treatment in kidney disease models. Lastly, the manuscript emphasizes the anti-inflammatory role of Tβ4 in inhibiting macrophage accumulation in the renal cortex, which may slow disease progression and reduce fibrosis.

Strengths:

The manuscript offers a well-organized and concise overview of Tβ4's role in kidney glomerulus, focusing on podocytes and macrophages. It thoroughly discusses relevant studies, highlighting the importance of maintaining podocyte cytoskeleton integrity for preserving glomerular function. Furthermore, the review explores the potential therapeutic applications of exogenous Tβ4 in glomerular disease, which could lead to novel treatments to reduce associated morbidity and mortality.

Suggestions for improvement:

The authors should address limitations in studying Tβ4's role in glomerular diseases, such as variations in animal models, experimental protocols, and limited data on human podocytes. Providing more context on Tβ4's relevance in human glomerular diseases, including its therapeutic potential and the need for further studies, would strengthen the review. Additionally, discussing Tβ4's potential role in other glomerular cell types or processes relevant to disease progression, such as endothelial cells, mesangial cells, and fibrosis, could provide a more comprehensive understanding of its impact on glomerular diseases.
